# When and Why Demands Reveal Their Challenging Potential during Change

**DOI:** 10.3390/ijerph182413076

**Published:** 2021-12-11

**Authors:** Charlotte Blum, Thomas Rigotti

**Affiliations:** 1Work, Organizational and Business Psychology, Johannes Gutenberg-University Mainz, 55122 Mainz, Germany; rigotti@uni-mainz.de; 2Leibniz Institute for Resilience Research, 55122 Mainz, Germany

**Keywords:** change, job demands–resources model, challenge–hindrance framework, change demands, workload, individual job impact, work engagement, autonomy, trust

## Abstract

This study contributes to a better understanding of the complexity of the demands that arise during organisational change. We investigated classic and change-specific demands in relation to emotional exhaustion and work engagement within the challenge–hindrance framework. We focused on workload and individual job impact and tested trust and autonomy as moderators. Data were collected via a cross-sectional online questionnaire. The convenience sample consisted of 388 employees experiencing organisational change at the time of responding. We conducted regression analyses to test for both direct and moderating effects. The results indicate that workload and individual job impact exhibit challenge and hindrance qualities. We also identified the significant moderating effects of trust and autonomy on individual job impact. This study integrates the challenge–hindrance framework into the job demands–resources model and offers a new perspective by applying this framework in the context of organisational change. We examined the specific roles of autonomy and trust regarding demands during change processes, and their potential in channelling challenging qualities is examined, offering new perspectives on the buffering of change demands.

## 1. Introduction

### 1.1. When and Why Demands Reveal Their Challenging Potential during Change

Staying profitable and competitive are permanent challenges for organisations that force them to evolve through change. Therefore, change has become increasingly common [1], and finding the key success factors of change has been a focus of research for some time [2,3]. Organisational change generates difficult situations in which employees may face dismissal, demotion, cancellation of financial incentives, and insecurity about their future job conditions. Changes in organisational culture, work colleagues, management teams, and other factors may also occur, which can negatively impact employees’ well-being [4,5]. Both researchers and operational experts agree that the reactions of the people affected by a change are essential determinants of the successful implementation of that change [1,2].

Change inevitably and unavoidably leads to additional demands on top of everyday challenges. Thus, it is important to better identify the demands that, during times of transformation, have an impact on employees’ emotional exhaustion and work engagement. One approach to understanding the effects of demands and resources within organisations is the job demands–resources model [6]. This model states that job resources and demands trigger two independent psychological processes. The first is a motivational process, in which job resources (e.g., autonomy, task variety, feedback, training and development, participation in decision-making, and supervisor support) have buffering effects on the stressor–strain relationship, leading to a positive motivational impact. There is consistent evidence that these job resources are positively associated with engagement [7]. The second psychological process, which was defined by Bakker and Demerouti [6], is a health-impairment process: job demands (e.g., workload, time pressure, role conflict, role ambiguity, emotional demands, physical demands, and work–family conflict [7,8]) are conceptualised as energy-depleting. When an individual’s job demands are excessive, sustained, or not balanced by adequate resources, their capacity may be overwhelmed, leading to strain and burnout [6].

Demands were originally thought to trigger only the health-impairment process. However, the challenge–hindrance framework suggests that certain demands, in addition to their resource-draining effects, may also enable personal development, learning and thriving, and increase motivation [9]. Regarding the conditions that may trigger the potential challenges of certain demands, research has so far focused on coping behaviour [10], job resources [11], and appraisal processes [12].

The context of change is innately demanding but has not been investigated specifically as a setting for the challenge–hindrance framework. Thus, in this study, we tested the boundary conditions for potentially challenging demands during change. This study contributes to the challenge–hindrance framework as an extension of the job demands–resources model. It also strengthens the literature on organisational change by examining two demands (i.e., workload and individual job impact), their effects within the context of change, and the development of new approaches to mastering the challenges posed by organisational change. We investigated the effects and dynamics of workload and individual job impact in the context of change, seeking to identify the extent to which someone is personally impacted by the current change process. 

As stated above, in the context of organisational change, employees inevitably face multiple new demands. Consequently, the provision of resources is even more important than in calm and steady times. Investigating change-related resources and their potential to trigger the challenging qualities of demands adds to the understanding within the literature of change dynamics. The resources we focus on are trust and autonomy, as potential moderators. We argue that resources help to promote the challenging potential of demands. Trust is a central variable influencing an organisation’s culture during change [13,14]. Meanwhile, autonomy provides a certain degree of freedom in decision-making and adapting to new demands [15], and was shown to have a positive impact on the way that people work. Sustaining employees’ involvement, engagement, and well-being is a crucial factor for the success of any organisational change initiative [16], both during and after the change. By embedding the job demands–resources model within the challenge–hindrance framework in the context of change, we can develop a better understanding of the interplay of demands and resources, which inevitably arises during transformation processes. Thus, in this study, we hypothesised that resources, trust, and autonomy trigger the challenging potential of workload and individual job impact within the context of change.

### 1.2. Challenge and Hindrance Demands in the Context of Change

Podsakoff et al. [9] were among the first to suggest that demands be classified as either challenge demands or hindrance demands. Employees are more willing to invest their energy, time, and resources into responding to demands that they perceive as challenging [7,17]. This particularly occurs when they feel confident that they can meet such demands successfully, and, if they make an invested effort, is likely to be recognised and rewarded [7]. Consequently, although demands may be energy-depleting, challenging demands also have a motivational quality that results in a positive affective state and increased work engagement [7].

Many studies showed that employee engagement was positively related to desirable outcomes such as organisational commitment, individual and group well-being, and organisational performance [7,18]. Engagement is defined as a positive, fulfilling, work-related state of mind characterised by vigour, dedication, and absorption [19]. Burnout, on the other hand, harms employee health and well-being, increases turnover and absenteeism, and negatively impacts job performance [20,21]. We will focus on emotional exhaustion, as the core symptom of burnout, which can be defined as “[…] a chronic state of physical and emotional depletion that results from excessive job demands and continuous hassles” [22] p. 486.

Crawford et al. [7] conducted a meta-analysis that operationalised challenging demands as a higher-order factor comprising job responsibility, time urgency, and workload. Hindrance demands were also operationalised as a higher-order factor consisting of situational constraints, role conflict, and role ambiguity. They found that all challenge demands were positively related to engagement and burnout, while hindrance demands were negatively associated with engagement and positively associated with burnout. Based on this, Crawford et al. [7] suggested that categorising demands into challenge and hindrance demands was important for understanding their differential effects.

While the challenge–hindrance framework is receiving increased attention, only a few challenge demands (e.g., workload, job responsibility, and time pressure) have been analysed to date. More research is needed to better understand the complex dynamics of job demands [23]. Organisational change inevitably leads to additional demands, which negatively impact the well-being of organisational members [4,5]. Analysing the challenge–hindrance framework within the context of change is an interesting avenue of inquiry, as it may help to identify how motivation might be fostered and negative strain effects reduced, ultimately supporting the successful implementation of change. 

### 1.3. Challenging Qualities of Demands

This study examined two demands and considered their potential to show challenging properties in the context of change: workload and personal job impact. Workload is a general but still relevant demand during change processes [24] and was previously shown to have a challenging potential. We also included personal job impact, as it is a change-specific demand.

**Workload**. The health-impairing qualities of workload have already been examined in the context of change [24]. Greenglas et al. [25], for example, identified the strong negative effect of nurses’ increased workload due to restructuring, leading to emotional exhaustion. Workload was repeatedly shown to have challenging qualities [7,8,23] outside of the context of change. However, the challenging potential of workload was not investigated in the context of change.

Change processes usually entail additional work for the people who are affected by them. During change processes, change-specific tasks (e.g., learning new IT skills or new workflows) need to be performed. These tasks are added to the employees’ everyday tasks, thereby increasing their workload. Change also leads to uncertainty, which people try to manage by gaining control. Workload is likely one of the demands that people can control; therefore, it appears to be more predictable than other variables that may arise during change. Spector [15] and Spector et al. [26] showed the positive effects of the locus of control on the well-being of employees. Therefore, we assumed that workload may function as a challenging demand. A clear understanding of when and how workload can become a challenge demand, rather than a hindering demand, in the straining context of change is a major asset during change implementation. To examine whether workload also functions as a challenge demand during times of transformation, we formulated two hypotheses:

**Hypothesis** **1.**
*The workload experienced during change is positively associated with emotional exhaustion.*


**Hypothesis** **2.**
*The workload experienced during change is positively associated with work engagement.*


**Individual Job Impact**. According to Caldwell et al. [27], individual job impact is defined as a change recipients’ assessment of the extent to which their job demands, expectations, and responsibilities are impacted by an organisational change. Job impact is related to both the way that employees respond to a change [27,28] and their level of change commitment [29]. This applies even when a person is positively predisposed toward a change or when the outcomes of a change are expected to be positive. Furthermore, studies have shown [30] that the more disruptive a change is, the more that uncertainty, fear of failure, and the loss of control increase. Caldwell et al. [27] claimed that circumstances of lower utility or lower valence outcomes lead to reduced effort, based on cognitive motivation models [31]. Accordingly, if individuals assume the change will have negative outcomes, they are less motivated to partake in change-related decision-making, and vice versa. Additionally, few studies on change considered the fact that change is perceived favourably by some people and unfavourably or neutrally by others. The schematic perspective by Lau and Woodman [32] offers such an approach, describing the effects of change as individually divergent depending on cognitive schemata. Perceived individual job impact may trigger both negative and positive consequences. Higher individual job impact indicates a greater personal involvement, and Fedor et al. [29] showed that individual job impact can have a positive relationship with change commitment if changes are less pronounced for the work-level unit. Therefore, in this study, we predicted that individual job impact might show features of challenge demands:

**Hypothesis** **3.**
*Individual job impact experienced during change shows a positive association with emotional exhaustion.*


**Hypothesis** **4.**
*Individual job impact experienced during change shows a positive association with work engagement.*


### 1.4. Channelling Challenging Potential

In this section, we argue that trust and autonomy trigger the challenging potential of workload and personal job impact by strengthening the positive relationship between demands and work engagement. In addition, we suggest that trust and autonomy buffer the positive relationship of demands with emotional exhaustion. 

**Trust**. Trust is defined as “a psychological state comprising the intention to accept vulnerability based upon positive expectations of the intentions or behaviour of another” [33] p. 395. This study focused on trust in management. Whether a leader is perceived as trustworthy depends on employees’ expectations about the leader’s future behaviour, meaning the expectation that their leader will not behave in ways that threaten their interests. Highly trusted leaders are expected to demonstrate competent behaviour and integrity and act in their employees’ best interests [33].

The theory of uncertainty management [34] states that, in times of instability and transformation, employees pay increasing attention to fairness, transparency, and trust. When an employee trusts that their organisation, leaders, and colleagues have their best interests in mind, they are much more willing to invest in the company and engage in change process [35,36]. Trust in one’s leader is an important enabler of successful change [37,38]. In a meta-analysis, Dirks and Ferrin [39] showed that trust in one’s leader is positively associated with organisational citizenship behaviour, job performance, job satisfaction, and organisational commitment. In change contexts, which are usually characterised by high levels of outcome uncertainty and ambiguity, trust is likely to be a key concern for people affected by the change, and thus a core determinant of their reactions. The definition of trust developed by Rousseau et al. [33] highlights the deeply relational nature of trust. Whether a person accepts vulnerability by another depends on their expectations of that other person. Hence, an employee’s acceptance of and commitment to a change process and its potential consequences relies on the anticipated behaviour and intentions of their leader. If such expectations are positive, the expectations regarding the change should also be positive. 

Thus, trust in leaders should trigger the challenging potential of demands during change. As explained, the job demands–resources model defined by Bakker and Demerouti [6] demonstrated the buffering effects of personal and organisational resources for demands on strain. Therefore, in this study, we hypothesised that trust buffers the stressor–strain path and enhances the challenge–work engagement path: 

**Hypothesis** **5.**
*Trust moderates the relationship between (a) workload and (b) individual job impact and work engagement, with higher levels of trust promoting these relationships.*


**Hypothesis** **6.**
*Trust moderates the positive relationship between (a) workload and (b) individual job impact and emotional exhaustion, with higher levels of trust reducing the effect.*


**Autonomy**. Autonomy was proven to be positively related to work engagement [7]. Intervention studies showed that modifications in job characteristics mediate the relationship between job redesign and employee well-being [40]. These findings inspired job-redesign interventions that sought to optimise the balance between job demands and resources. Creating resourceful work environments is becoming increasingly important when considering organisational changes, which may involve technological advances and new forms of work [23]. Schaufeli et al. [41] conducted a two-wave longitudinal study of managers and proved that changes to job resources contribute to changes in employee well-being. Specifically, they found that an increase in social support, autonomy, opportunities to learn, and performance feedback resulted in higher levels of work engagement one year later.

Autonomy is a well-examined resource that was proven to have a positive impact on well-being and motivation [2,42,43]. This is especially relevant in changing working contexts. Situations of transformation create non-negotiable circumstances and demands for the people involved. For example, if the management of a company decides to implement a new IT system, this must be accepted by all employees. However, offering degrees of autonomy in the way the system is introduced or implemented may function as a resource. In this study, autonomy is regarded as a structural resource. As job resources can buffer the negative effects of job demands and increase motivation, they are likely to also play a role in unfolding the challenging potential of demands. As explained above, employees who feel confident that they can meet a demand will invest more of their time, energy, and motivation in doing so [7]. When employees can choose how they approach new tasks and solve problems in their own way, this increases their confidence and motivation. Consequently, we hypothesised that autonomy promotes the challenging potential of certain demands, resulting in positive affective states, such as work engagement [7]. To examine the impact of autonomy on demands and their challenging potential during change processes, as well as the potential buffer effect on the relationship between demands and strain, we developed the following hypotheses:

**Hypothesis** **7.**
*Autonomy moderates the positive association between (a) workload and (b) individual job impact and work engagement, with higher levels of autonomy promoting these relationships.*


**Hypothesis** **8.**
*Autonomy moderates the positive association between (a) workload and (b) individual job impact and emotional exhaustion, with higher levels of autonomy lessening the strength of these relationships.*


A summary of the conceptual research model for this study is shown in Figure 1.

## 2. Materials and Methods

The data collection took place in 2018 via an online questionnaire that was distributed via our private and business contacts, alumni networks, and social networking websites. The criterion for participation was involvement in a change process at the time of answering the questionnaire. We do not know how many potential respondents received an invitation to the study, and thus we cannot report response rates. The study included neither manipulation nor the assessment of extensive, private, or sensitive data, and data were processed completely anonymously throughout the process. The participants were informed about the goals of the project and how their data would be stored and processed. We included an informed consent question at the start of the questionnaire, which the respondents had to actively agree to in order to continue with the questionnaire. The data collection and processing approach complied with the European data protection legislation. The setting of change was continuously used as a point of reference for all questions within the questionnaire. The questionnaire asked about general and change-specific job demands, resources, leadership styles, and well-being, as well as demographic information and the characteristics of participants’ workplaces and change processes. Participation was anonymous, voluntary, and uncompensated. Participants were offered the option of receiving an aggregated report of the research results at the end of the project by e-mail. Their email addresses were stored separately from the questionnaire responses.

### 2.1. Sample

Overall, 388 participants were recruited, of which 260 were female (76.0%) and 128 were male (33.0%). Their ages ranged from 18 to 65 years, with *n* = 154 (39.7%) being between 18 and 30 years, *n* = 155 (39.9%) between 31 and 40 years, *n* = 52 (13.4%) between 41 and 50, and 27 (7.0%) between 51 and 65 years old. Most participants had a university-entrance qualification (*n* = 326, 84.0%), with 295 (76.0%) having a general university degree. Participants had worked for their companies for an average of 5.84 years (SD = 5.86). The sample consisted of 156 persons with leadership responsibilities and 235 people without leadership roles.

The companies where the participants worked covered a wide range of industries, including the automotive, e-commerce, finance, and pharmaceutical sectors. The most common of the 32 categories was logistics and transport (*n* = 66, 17.0%). When asked about the type of change they were experiencing, 63.7% (*n* = 247) said they were experiencing internal restructuring, 30.9% (*n* = 120) a business expansion, 21.9% (*n* = 85) staff reductions, 12.6% (*n* = 49) a merger or acquisition, 11.1% (*n* = 43) a change of location, and 11.1% (*n* = 43) outsourcing. Almost 25% (24.7%, *n* = 96) answered “other”. For 210 participants (54.1%), their area of responsibility and type of tasks had changed, while 228 (58.8%) reported changes in their work processes, 79.9% (*n* = 310) changes in their team, 226 reported (58.2%) new technical equipment such as software or machines, 173 reported (44.6%) new products or services, 139 reported (35.9%) a new manager, and 101 reported (26.0%) new tasks. The questionnaire also asked whether the participants had experienced additional changes that were not referred to in the survey, which 45 (11–6%) participants affirmed, naming changes such as the creation of a new business within their company, the appointment of a new CEO, their promotion to a management position, and the adoption of a new working culture (e.g., flexible working hours, remote work, an open-plan office).

### 2.2. Measures

All the questions were introduced with a reminder that the answers needed to refer to the current change process. Additionally, all responses were measured using a 5-point Likert scale. If not otherwise stated, the response options ranged from ‘does not apply at all’ (1) to ‘largely applies’ (5).

### 2.3. Individual Job Impact

Individual job impact was measured with a seven-item scale developed by Caldwell et al. [27] that was back-translated. It included statements such as, “Due to the change, I feel more pressure at work” or “My area of responsibility at work has changed”. Cronbach’s α was 0.86.

### 2.4. Autonomy

Autonomy was measured using a 5-item scale developed by Rosenthal et al. [44], plus one additional item to account for the changed situation: “I can decide how to execute the change”. Other statements included, “I choose my tasks” or “I can influence the organisation of my tasks”. Cronbach’s α was 0.85.

### 2.5. Workload

Workload was measured with a 5-item scale from the third version of the Copenhagen Psychosocial Questionnaire (COPSOQ III; [45]), including items such as “Do you have to work very fast?” and “Do you get behind with your work?” The response options ranged from ‘very seldom/never’ (1) to “very often/always” (5). Cronbach’s α showed satisfactory reliability of 0.80.

### 2.6. Trust

Trust in management was assessed with three items, including “To what extent do you trust senior management to look after your best interests?” [46]. Cronbach’s α was 0.84, proving good reliability.

### 2.7. Work Engagement

Work engagement was assessed with six items [47], including three each for the facets of vigour and dedication. Absorption was omitted due to its state-like character. The items included “My job inspires me” and “I am proud of the work that I do”’. The scale showed a good reliability (Cronbach’s α = 0.91).

### 2.8. Emotional Exhaustion

Three items from the Maslach Burnout Inventory [48] were used to measure exhaustion: “I feel emotionally drained from my work”, “I feel fatigued when I get up in the morning and have to face another day in the job”, and “I feel burned out from my work”. The scale had good reliability (Cronbach’s α = 0.88).

### 2.9. Confirmatory Factor Analyses

We conducted confirmatory factor analyses to test the conceptual distinctness of the measures using Mplus version 7.3 [49]. The hypothesised model consisted of six factors: workload, individual job impact, trust, autonomy, work engagement, and emotional exhaustion. The results for the six-factor model yielded acceptable fit statistics: χ^2^ = 814.39, df = 309, comparative fit index (CFI) = 0.90, Tucker–Lewis index (TLI) = 0.88, and root-mean-square error of approximation (RMSEA) = 0.07. The proposed model was then compared to a five-factor model combining workload and individual job impact (χ^2^ = 1185.45, df = 314, CFI = 0.82, TLI = 0.80, RMSEA = 0.09) and a one-factor model (χ^2^ = 3243.44, df = 324, CFI = 0.40, TLI = 0.35, RMSEA = 0.16). Based on these indices, the hypothesised six-factor model was chosen as the final model for this study.

## 3. Results

### 3.1. Descriptive Statistics

We found that emotional exhaustion was strongly positively correlated with workload, individual job impact, and trust, and strongly negatively correlated with work engagement and autonomy (see Table 1 for detailed values). Work engagement showed no significant correlation with individual job impact or workload but was strongly correlated with trust and autonomy (Table 1).

### 3.2. Main Effects

Hypotheses 1 to 4 predicted positive relationships between personal job impact, workload, and work engagement, as well as with emotional exhaustion. Workload showed no significant association with work engagement (*r* = 0.09, *p* = 0.106), but was significantly associated with exhaustion (*r* = 0.37, *p* < 0.001). Likewise, personal job impact was not found to be significantly related to work engagement (*r* = 0.10, *p* = 0.081), but was significantly related to emotional exhaustion (*r* = 0.24, *p* < 0.001). However, in the multiple moderated regression analyses (see Table 2), both demands showed a direct positive relationship with work engagement and emotional exhaustion, except for the model including autonomy in the case of personal job impact. Therefore, hypotheses 1 to 4 were mostly supported by the data.

### 3.3. Moderation Analysis

To test hypothesis 5 (i.e., “Trust strengthens the association between (a) workload and (b) individual job impact and work engagement”), a linear moderated regression was conducted. All predictor variables were mean-centred. We found that trust had a significant interaction with individual job impact (β = 0.12, *p* = 0.017), but not with workload in relation to work engagement, supporting H5b but not H5a. The significant interaction is plotted in Figure 2. Simple-slope analyses revealed a significant positive association between individual job impact and work engagement for high trust (β = 0.24, *p* < 0.001), but not for low trust (β = 0.01, *p* = 0.846).

We also conducted a linear-moderated regression to test hypothesis 6 (i.e., “Trust weakens the positive association between (a) workload and (b) individual job impact and emotional exhaustion”). Trust did not have a significant moderating effect on workload (β = 0.04, *p* = 0.443) or individual job impact (β = −0.08, *p* = 0.139; Table 2) in relation to emotional exhaustion. Hence, H6a and H6b were not supported. 

Hypothesis 7 (i.e., “Autonomy strengthens the relationship between (a) workload and (b) individual job impact and work engagement”) was tested using a moderated regression analysis. As shown in Figure 3, a positive moderating effect on the relationship between individual job impact and work engagement was found (β = 0.13, *p* = 0.012). Simple-slope analyses showed a significant positive relationship between individual job impact and work engagement for high autonomy (β = 0.20, *p* = 0.003), but no significance for low autonomy (β = −0.05, *p* = 0.421). These results support H7b, but not H7a.

Finally, we tested hypothesis 8 (i.e., “Autonomy weakens the positive association between (a) workload and (b) individual job impact and emotional exhaustion”). No significant moderating effects were found for either individual job impact (β = 0.06, *p* = 0.268; Table 2) or workload (β = 0.01, *p* = 0.889; Table 2). Hence, H8a and H8b were disproven. Both autonomy and trust had a significant direct negative relationship with emotional exhaustion.

## 4. Discussion

The purpose of this study was to investigate the conditions under which the challenging potential of workload and personal job impact unfolded within the context of organisational change. As trust and autonomy showed theoretically positive relevance during change, they were examined as moderators in the relationship between demands and work engagement, as well as emotional exhaustion. The results demonstrate that workload and personal job impact have both hindering and challenging potential, even under circumstances of change. Personal job impact was found to have challenging characteristics under conditions of high trust and autonomy. No moderation effects were found for workload in relation to emotional exhaustion or work engagement. Importantly, no moderation effects were found regarding emotional exhaustion. Therefore, although autonomy and trust boost motivational potential, neither of them mitigate the straining effects of workload and job impact.

### 4.1. Workload and Individual Job Impact as Potential Challenging Demands

Workers’ motivation and well-being are influenced by their working conditions [50]. The job demands–resources model provides a well-validated framework for understanding employee motivation and well-being. Research on the challenge–hindrance framework supported the differentiation of challenging and hindering demands [17,51]. Change is demanding for employees; thus, it is critical to understand the conditions under which demands reveal their challenging potential. The results of this study demonstrate that, in the context of change, workload has both hindering and challenging potential. While workload was found to be positively related to emotional exhaustion, it also had a positive relationship with work engagement, at least under the control of resources. Neither trust nor autonomy was shown to enhance the challenging character of workload. Future research could further explore the factors that strengthen the relationship between workload and work engagement, especially during change. This study assumed that workload was one of the most common demands during periods of organisational change. Identifying ways to buffer the negative consequences and strengthen the positive consequences of workload would be very valuable.

Individual job impact was positively related to both work engagement and emotional exhaustion. Therefore, individual job impact did demonstrate challenging potential during change. In addition, both trust and autonomy were proven to enhance the challenging potential of individual job impact. This suggests that factors such as working conditions and organisational culture can have a major impact on the effects of change.

Both trust and autonomy had highly significant positive relationships with work engagement and highly significant negative relationships with emotional exhaustion. This indicated that both factors strongly support the implementation of change and can enhance the challenging potential of individual job impact.

### 4.2. Theoretical Contribution

In addition to the uncertainty management theory [34], the job demands–resources model [6] and the challenge–hindrance framework [51] served as theoretical frameworks for this study. This study offers a new perspective on these theories, as it operationalises individual job impact [27] as a challenge demand. This fills a significant gap in the existing research on change [31,52]. As Caldwell et al. [27] stated, the incorporation of this perspective is vital for a better understanding of change implementation.

Trust, and its role as a moderator, is also a valuable addition to the model. Trust plays a vital role in affective dynamics in organisations and change processes. As an organisational resource that triggers challenging potential, trust adds to the research on both the job demands–resources model and the challenge–hindrance framework. Zak et al. [14] demonstrated the reciprocal nature of trust, meaning that the behaviour of people affected by a change is influenced by those around them; this impacts their willingness to engage in the change process. This study showed that, under conditions of high trust, the relationship between individual job impact and work engagement can be positive. This illustrates that trust triggers the challenging potential of demands during change, thereby adding to the literature on trust as an organisational resource and on change management more broadly.

Autonomy is a well-established organisational resource within the job demands–resources model; however, before this study, it had not been linked to the context of change, nor to the challenge–hindrance framework specifically. This study demonstrated that autonomy functions as an organisational resource during periods of change, triggering the challenging potential of individual job impact. 

### 4.3. Limitations and Future Directions

There were limitations to the interpretation and generalisability of the results in this study. Since a cross-sectional design was used, it was not possible to draw causal conclusions. Another limitation was the operationalisation of work engagement. Only two (i.e., vigour and dedication) of the three dimensions of the original scale by Schaufeli et al. [47] were chosen; absorption was excluded. 

The participants were required to be involved in a change process at the time of the questionnaire. Those processes differed in type and timing, which might have biased the results. Future research should investigate specific change scenarios to reveal matching challenging demands. 

Future research should examine further personal resources, including self-efficacy, optimism, and self-confidence, because such resources promote individuals’ perceptions of whether they can cope with a demand and were proven to have positive effects on well-being [27,53,54]. Another interesting factor is motivation, which concerns the change process itself. The importance of change commitment during change processes was proven [16,55]; however, its role within the challenge–hindrance framework during change was not investigated. Therefore, future research could explore the possible moderating effects of personal resources and change commitment in supporting the challenging potentials of demands during change.

Furthermore, change fairness [56,57] and organisational justice [34,50,58] are significant concepts in the context of change, both alone and in combination with the challenge–hindrance framework. These factors can mitigate great uncertainty, potential threats, and a lack of information [2,59] due to change. Since this study found that trust showed positive qualities, it would also be valuable to investigate the role of such factors regarding trust.

Finally, it would be useful to examine change processes on a longitudinal basis to achieve a better understanding of intraindividual dynamics. Kuppens et al. [60] showed that the more an individual’s affective experience fluctuates over time, the more likely they are to experience decreased well-being and adjustment. While individual job impact captures the personal perception of change, future research could investigate intra-individual dynamics in the context of change in the workplace.

### 4.4. Practical Implications

The findings of the present study offer several practical implications that will support employers in implementing organisational change more successfully and strengthen their employees’ well-being during change. Firstly, working actively with the challenge–hindrance framework during times of transformation is a promising approach, given that additional demands inevitably arise at such times. Identifying ways for demands to trigger positive outcomes would benefit organisations that are undergoing change. Understanding that demands may not only harm employees’ well-being but may also have a challenging potential is highly strategically valuable. Analysing organisation-specific demands will enable leaders and organisations to better manage them. Accordingly, organisations should (a) analyse the resources already present in their organisation that could buffer and/or channel challenging potential during their change process or (b) identify the resources that might work well for their specific context, which have not yet been implemented. Based on these analyses, organisational resources could be developed to support the implementation of change. For both demands and resources, it is advisable to consider organisational and personal factors. Generally, transparent communication during organisational change results in a higher level of trust for management [61]. This study shows that trust may promote the challenge potential of change demands. Hence, employers should seek to establish an organisational culture that promotes trust, particularly during change.

Lastly, autonomy has promise as a structural resource during change, as it was proven to trigger challenge potential for the individual experience of change. Allowing employees to analyse, manage and shape their demands gives them a sense of freedom and control over the change. This promotes positive outcomes for their well-being. An individual that is confronted with new demands will feel more self-sufficient and in control when they are empowered to decide how to approach and solve those demands. When planning the process and implementation of change, employers should consider which tasks could be dealt with by the employees autonomously. The possibility of finding individual solutions should also be clearly communicated.

## 5. Conclusions

This study compared the change-specific demand of individual job impact and the non-change-specific demand of workload in terms of challenging and hindering potential. It identified significant effects on both work engagement and emotional exhaustion. Autonomy and trust further triggered the challenging potential of personal job impact, but not of workload. 

As change is a context in which additional demands inevitably arise, it is vital to better understand the factors that trigger potential challenge demands and support challenging dynamics. Understanding these patterns enables the minimisation of hindrance effects and supports employees’ motivation and engagement in change processes. As both autonomy and trust were found to be moderators of the relationship between individual job impact and work engagement, they should be promoted as organisational resources through the development of appropriate organisational cultures. Employers should undertake analyses of resources and demands at an organisational and personal level. Organisational and leadership practices that enable the active promotion of resources and shape demands (e.g., through job crafting as well as more transparent and fairer communication) positively influence the change process. Further investigation of additional challenge–hindrance demands is required to better understand and support change processes.

## Figures and Tables

**Figure 1 ijerph-18-13076-f001:**
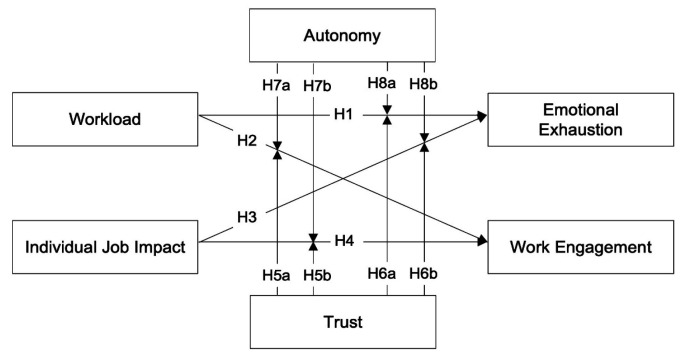
Conceptual Research Model.

**Figure 2 ijerph-18-13076-f002:**
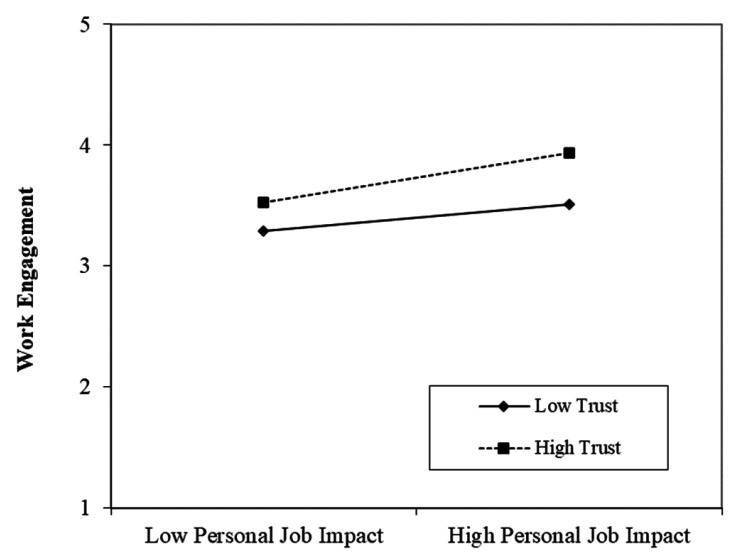
Relationship between individual job impact and work engagement moderated by trust.

**Figure 3 ijerph-18-13076-f003:**
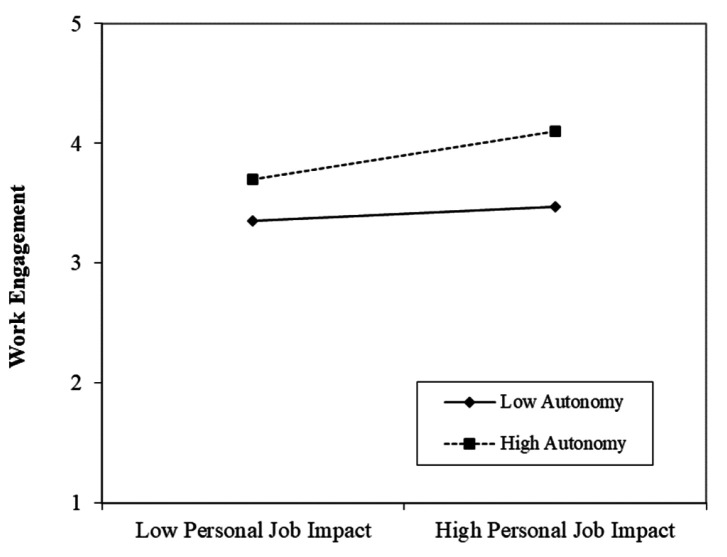
Relationship between individual job impact and work engagement moderated by autonomy.

**Table 1 ijerph-18-13076-t001:** Means, standard deviations and correlations of study variables.

Variable	*M*	*SD*	1	2	3	4	5	6
1. Emotional exhaustion	2.26	0.97	–					
2. Work engagement	3.42	0.83	−0.48 **	–				
3. Workload	3.41	0.77	0.37 **	0.09	–			
4. Individual job impact	3.09	0.84	0.24 **	0.10	0.43 **	–		
5. Trust	3.37	1.05	−0.33 **	0.41 **	−0.12 *	−0.09	–	
6. Autonomy	3.50	0.74	−0.25 **	0.45 **	−0.07	0.04	0.32 **	–

*Note.* ** *p* < 0.001, * *p* < 0.05.

**Table 2 ijerph-18-13076-t002:** Moderator analysis: effects of all variables on work engagement and emotional exhaustion.

	Work Engagement			Emotional Exhaustion		
	B(SE)	β	95% CI [LL, UL]	B(SE)	β	95% CI [LL, UL]
Intercept	3.41(0.04)			2.26(0.05)		
Personal Job Impact	0.07(0.05)	0.08	[−0.02; 0.17]	0.30(0.06)	0.26 ***	[0.17; 0.42]
Autonomy	0.49(0.06)	0.43 ***	[0.37; 0.60]	−0.038(0.07)	−0.028 ***	[−0.52; −0.23]
Personal Job Impact × Autonomy	0.17(0.07)	0.13 *	[0.04; 0.30]	0.09(0.08)	0.06	[−0.07; 0.25]
Intercept	3.40(0.04)			2.27(0.05)		
Personal Job Impact	0.13(0.05)	0.14 **	[0.03; 0.23]	0.26(0.06)	0.22 ***	[0.14; 0.38]
Trust	0.33(0.04)	0.42 ***	[0.25; 0.41]	−0.29(0.05)	−0.31 ***	[−0.39; −0.19]
Personal Job Impact × Trust	0.11(0.05)	0.12 *	[0.02; 0.20]	−0.08(0.06)	−0.08	[−0.19; 0.03]
Intercept	3.42(0.04)			2.27(0.05)		
Workload	0.12(0.06)	0.11 *	[0.01; 0.24]	0.47(0.07)	0.36 ***	[0.34; 0.60]
Autonomy	0.52(0.06)	0.45 ***	[0.40; 0.63]	−0.32(0.07)	−0.24 ***	[−0.46; −0.18]
Workload × Autonomy	0.11(0.07)	0.08	[−0.03; 0.25]	0.01(0.09)	0.01	[−0.16; 0.18]
Intercept	3.41(0.04)			2.28(0.05)		
Workload	0.18(0.06)	0.16 **	[0.06; 0.29]	0.44(0.07)	0.33 ***	[0.30; 0.57]
Trust	0.34(0.94)	0.43 ***	[0.26; 0.42]	−0.27(0.05)	−0.28 ***	[−0.36; −0.17]
Workload × Trust	0.04(0.06)	0.04	[−0.07; 0.15]	−0.002(0.06)	−0.01	[−0.14; 0.11]

*Note.* B = unstandardised coefficient; SE = standard error; β = standardised coefficient; total *n* ranges from 307 to 311; LL and UL = the lower and upper limits of a confidence interval (CI), respectively; * *p* < 0.05, ** *p* < 0.01, *** *p* < 0.001.

## Data Availability

The data presented in this study are available on request from the corresponding author.

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
