# Peer review of "When and Why Demands Reveal Their Challenging Potential during Change"

_ijerph, 2021, doi:10.3390/ijerph182413076_

Round 1

Reviewer 1 Report

An organisational change or change processes do not necessarily imply an increased workload. Perhaps it is the acquisition of new skills or just knowledge on how to deal with the new procedure that might require more time. The section on workload is introduced in a concise way. The article would benefit from a more detailed description of the workload rationale as an antecedent, providing more references. Why didn't you use Spector et al., 2002 to justify workload as antecedent?

It is not immediately clear that you are considering trust in the leader. After the description of trust it would be useful to introduce why trust at the relational level is critical and then, especially when considering the leader. "In other words" (216) sounds strange immediately after the explanation of trust in (anyone).

Please provide the type of description shared with participants in the survey on what organisational change should be considered while answering the questionnaire.

It might also be useful to see more items and get a proper idea of the dimensions investigated, as participants responded by considering different change scenarios (e.g., staff reduction or change of location), each of which implies different consequences in terms of workload, trust in management, autonomy, etc.

Have you considered comparing your results according to the different types of organisational change faced by participants? Or at least some of them that can be differentiated in terms of workload and impact (perhaps grouping them on this two levels of analysis)? Just a thought...

Please provide the time frame of your data collection.

It would be wise to change the way the authors discuss the results for two main reasons. Firstly, this is a cross-sectional study (as stated in the limitations) and secondly, it is not clear where and if the various changes were successfully implemented. Therefore, I would use a more probabilistic expression when discussing the theoretical and practical implications.

The practical implications are confusing and the line of reasoning behind them is not always clear. Little is written about how organisations and managers could benefit from these results in terms of building trust in the change and in their leaders and how to leave autonomy to people to manage their own impact of the change on their job. 

Please remove "Au" at lines 453 

Reviewer 2 Report

This study focuses on the role of workload and job impact in explaining  exhaustion and engagement in the context of organizational change. It also includes potential moderators of the hypothesised relationships (i.e., trust and autonomy). The topic of the study is interesting and the paper is generally well-written. Here are my concerns:

  1. Please give more information on the study sample in the abstract.
  2. In the first part of the Intro (p. 2 line 73 and again p. 2 line 95) it is reported for the first time the notion of “job impact” but nothing is said about what this factor actually is, leading to some disorientation in the reader. Please add a sentence (or a couple of words) to clarify.
  3. 4 line 143, please leave out the notion of “dynamics”, since the study design does not permit to address dynamics.
  4. 4 from line 166. The paragraph on job impact is really very long. I would try, if possible, to shorten it a bit. Honestly, I’m not particularly convinced by the arguments provided to justify job impact as a challenge demand, also because the presented review of the literature seems to suggest that job impact is actually a hindrance. I’m wondering whether such arguments can be made stronger and based on previous evidence.
  5. Similarly, the paragraph regarding trust (p. 5 and 6) is really very long. Again, I would try to shorten this part too to facilitate the reading.
  6. Method section. How many questionnaires were distributed initially? Giving an idea of the response rate would be helpful.
  7. What about informed consent? Was there any procedure in place to this end?
  8. The age classes reported in the “Sample” section (p. 7) are not mutually exclusive. Please check and amend.
  9. In my view, the “Analysis” section (end of p. 8) should explain the type of analyses authors are going to run, not the results. Here, I would also add info on the analyses with which the main hypotheses will be tested (i.e., multiple regression).
  10. End of p. 8, the chi-square symbol cannot be read, please check.
  11. Table I. How is it possible that trust has a positive relationship with emotional exhaustion (i.e., .33)? Please check and if this is not a typo, I think explanations should be provided.
  12. 10, pararaph from line 428. There is a mistake in my view: trust should not “strengthen” the relationship between workload and job impact and burnout. Trust weakens such relationships according to the formulated hypotheses.
  13. Same page (i.e, 10) please check the sentence from line 432 to 435. It doesn’t read well in my view.
  14. Similarly, check the sentence from line 436 to 438.

In general, this is a good study. If possible, to increase readability (and the possibility to be quoted), I would strongly consider to shorten it a bit.

Reviewer 3 Report

Dear Authors, the text is well written. The literature review is not objectionable. The research process was well planned and conducted. The research findings and conclusions are interesting. I believe that the article is suitable for publication in this form.

Round 2

Reviewer 2 Report

Dear authors,

thank you for incorporating my suggestions in the revision. I have few remaining issues, very minor in nature. Overall I think this is a interesting contribution to the literature, especially as far as the notion of job impact and its double nature (i.e., challenge and hindrance) is concerned. Of course the cross-sectional design of the study invites caution. I hope you will have the opportunity to rely on a longitudinal study in the future. 

1-Please amend the last sentence in the abstract, 'offer' should become 'offering'

2-Please also check all the paper for typos. I found others in addition to the one reported in comment 1.

3-Line 58: "When it comes to the question of which conditions the potential challenges of certain demands may unfold". Perhaps better "When it comes to the question of which conditions may unfold the potential challenges of certain demands".

4-line 125. Please provide a reference here: "With workload, we chose a general but still relevant demand during change procesess,..."

5-line 345. The chi square associated to the one factor model (0.09) is not correct. Please report the correct one.

6-Table 2. The quality (format) of the table should be improved. At present it is not very easy to read and understand all the reported data (e.g., some confidence intervals are part in one line and part in another line).

7-The figure representing the conceptual model (currently on p. 3) should perhaps be moved below, just before the Method section. This is because the figure mentions the hypotheses, which however have not been already discussed on p. 2. Please consider this suggestion.

Author Response

Responses to the Reviewers’ 2, Round 2 comments

Dear Editor(s), and Reviewer 2.

    Thank you very much for your efforts and valuable feedback in this second round. Based on your feedback, we were able to dissolve flaws in the manuscript and to finalize and improve the study even further regarding presentation and precision. Please find a point by point answer to each comment. We are looking forward to receive your feedback.

Best regards,

The authors

  1. Please amend the last sentence in the abstract, 'offer' should become 'offering'
    • Thank you for pointing this out, we have changed ‘offer’ to ‘offering’.
  2. Please also check all the paper for typos. I found others in addition to the one reported in comment 1.
    • Thank you again, we did check the manuscript again and should have found and corrected all typos.
  3. Line 58: "When it comes to the question of which conditions the potential challenges of certain demands may unfold". Perhaps better "When it comes to the question of which conditions may unfold the potential challenges of certain demands".
    • We have adapted the phrase accordingly.
  4. line 125. Please provide a reference here: "With workload, we chose a general but still relevant demand during change procesess,..."
    • We have included the reference of Balducci et al. (2021).
  5. line 345. The chi square associated to the one factor model (0.09) is not correct. Please report the correct one.
    • Thank you for pointing this out, we have the corrected the chi square accordingly.
  6. Table 2. The quality (format) of the table should be improved. At present it is not very easy to read and understand all the reported data (e.g., some confidence intervals are part in one line and part in another line).
    • We have improved the format to make it more readable.
  7. The figure representing the conceptual model (currently on p. 3) should perhaps be moved below, just before the Method section. This is because the figure mentions the hypotheses, which however have not been already discussed on p. 2. Please consider this suggestion.
    • Thank you for this suggestions. We have moved the table, where indeed it is better placed.